# Stratospheric Night Sky Imaging Payload for Space Situational Awareness (SSA)

**DOI:** 10.3390/s23146595

**Published:** 2023-07-21

**Authors:** Perushan Kunalakantha, Andrea Vallecillo Baires, Siddharth Dave, Ryan Clark, Gabriel Chianelli, Regina S. K. Lee

**Affiliations:** Department of Earth and Space Science and Engineering, York University, Toronto, ON M3J 1P3, Canada; andreavb@yorku.ca (A.V.B.); sgbdave@gmail.com (S.D.); rwclark@my.yorku.ca (R.C.); gabechi@my.yorku.ca (G.C.); reginal@yorku.ca (R.S.K.L.)

**Keywords:** space situational awareness (SSA), resident space objects (RSOs), commercial off-the-shelf (COTS), stratospheric balloon, star tracker

## Abstract

Space situational awareness (SSA) refers to collecting, analyzing, and keeping track of detailed knowledge of resident space objects (RSOs) in the space environment. With the rapidly increasing number of objects in space, the need for SSA grows as well. Traditional methods rely heavily on imaging RSOs from large, narrow field-of-view (FOV), ground-based telescopes. This research outlines the technology demonstration payload, Resident Space Object Near-space Astrometric Research (RSONAR)—a star tracker-like, wide FOV camera combined with commercial off-the-shelf (COTS) hardware to image RSOs from the stratosphere, overcoming the disadvantages of ground-based observations. The hardware components and software algorithm are described and evaluated. The eligibility of the payload for SSA is proven by the image processing algorithms, which detect the RSOs in the images captured during flight and the survival of the COTS components in the near-space environment. The payload features a low-resolution, wide FOV camera coupled with a Field Programmable Gate Array (FPGA)-based platform that houses the altitude and time-based image capture algorithm. The newly developed payload in a 2U-CubeSat form factor was flown as a space-ready payload on the CSA/CNES stratospheric balloon research platform to carry out algorithm and functionality tests in August 2022.

## 1. Introduction

The space environment has seen a rapid increase in the number of resident space objects (RSOs), ranging from tiny flecks of paint to entire satellites. In recent years, the space industry has trended toward mega-constellations of smaller satellites, further increasing the population of the space environment. RSOs can reach orbital speeds of over 7 km/s, traveling fast enough to cause devastating damage in collision events. Though once believed to be unlikely, the rapidly increasing RSO population is increasing the chances of in-orbit collisions. Such an event has already been realized in February 2009, when Iridium-33, an American communications satellite, collided with Kosmos2251, a defunct Russian military satellite, generating almost 2000 fragments with diameters of at least 10 cm, and thousands more smaller pieces [1].

With the threat of future in-orbit collisions growing as the number of RSOs increases, satellites must be tasked with collision avoidance maneuvers to prevent impacts. To develop such strategies, operators must first be aware of nearby RSOs, having detailed information such as the type of object and orbital parameters. Space situational awareness (SSA) refers to various programs and initiatives to advance such knowledge and can be accomplished by imaging RSOs and applying image processing algorithms to extract the required information about the objects.

In addition to optical imaging, radar systems have been used extensively for SSA activities. One example of such a system is the Italian Bi-Static Radar for LEO Tracking (BIRALET) system [2]. This radar system’s transmitter works in the P-band, operating at 410–415 MHz, able to capture low Earth orbit (LEO) RSOs in beam parking mode or tracking mode. Another example is the recently developed CHEIA SST Radar, a Romanian system leveraging existing parabolic antennas, aimed at strengthening the European Union Space Surveillance and Tracking (EU SST) effort [3]. This C-band radar system operates in the 3.6–6.4 GHz range, also able to track RSOs.

Many techniques have been developed to collect and process radar data, obtaining information critical to SSA. At Fraunhofer Institute for High Frequency Physics and Radar Techniques FHR, a team of scientists operate a radar system, the Tracking and Imaging Radar (TIRA) [4]. Information such as orbital elements, motion parameters, target shape, and mass and material properties have been gathered from this system. An initial orbit determination (IOD) algorithm has been developed to determine the state vector and orbital track of satellites using the Italian Bistatic Radar for LEO Survey (BIRALES) system [5]. The algorithm is being further developed to improve catalog correlation. Using radar data alongside optical data, a concept referred to as data fusion, is a promising technique for improving the information gained from RSO detections [6].

In terms of optical systems, traditionally, SSA activities typically focus on imaging RSOs from ground-based telescopes, such as NASA’s Michigan Orbital Debris Survey Telescope (MODEST) [7]. Such a large-scale, narrow field-of-view (FOV) telescope can take high-resolution images of RSOs in geosynchronous Earth orbit (GEO) [8]. The narrow FOV telescopes have a large integration time, for example, 5 s is typical, integrated with an active tracking methodology. The light from an object is accumulated in a narrow spatial region of the image, improving the sensitivity of the instrument. Active tracking is suitable for GEO objects, which move relatively slowly and more predictably with respect to a ground-based observer, making them easier to keep in the FOV of a telescope. However, the tracking requirements for a LEO observation introduce complications in the tracking system, since RSOs in LEO tend to move much faster than RSOs in GEO and have more uncertainty in their trajectories due to effects such as atmospheric drag. Furthermore, the narrow FOV would limit the number of RSOs that can be seen in a single image. Coupled with large design and operating costs, a need exists for a low-cost alternative with space-based operation, short integration time, and a wide FOV imager. Such an imager would also be useful in LEO RSO detections [9,10]. The imaging strategy proposed in this paper is designed to survey many RSO with a wide FOV, that is cost-effective and suitable for both ground and space applications [11].

In comparison to narrow FOV imagers, RSO detection and tracking using a wide FOV imager has been studied with limited results in the past. Clark et al. developed a simulator to realistically simulate images taken by low-resolution, space-based imagers [12]. Clemens demonstrated the feasibility of RSO detection from a wide FOV imager using images from the Cascade SmallSat and Ionospheric Polar Explorer (CASSIOPE) satellite’s Fast Auroral Imager (FAI) [13]. Dave et al. then examined an in-orbit RSO orbit estimation method using star tracker cameras [14]. Sease et al. predicted RSO motion from a simulated star tracker imager, demonstrating that angular rates can be extracted from streaked imagery without the use of star catalogs, and even extend image processing to detect RSOs [15]. Spiller et al. also examined RSO detection from star-tracker images using simulated datasets [16]. Wide FOV telescopes have also been considered for ground-based RSO imaging and subsequent image processing, as demonstrated by Hasenohr [17], and extended to wide FOV ground-based arrays by Fitzgerald et al. [18].

In this paper, we present the overview of the Resident Space Object Near-space Astrometric Research (RSONAR) payload with a wide FOV star tracker-like camera with relatively low resolution, built with commercial-off-the-shelf (COTS) components. The payload was flown on a stratospheric balloon as part of the Canadian Space Agency’s (CSA) stratospheric balloon program (STRATOS). There have been numerous remote sensing and technology demonstration missions onboard stratospheric balloons, carrying out experiments at sub-orbital altitudes as test missions for orbital demonstrations. Examples of such experiments include University of Alberta’s Ex-Alta 2 mission, designed to monitor wildfires with a multispectral imager [19], and University of Minnesota’s SOCRATES payload, studying x-rays and gamma rays [20]. To the best of our knowledge, the SSA mission described in this paper was the first mission of this nature with the ambitious goal to image night sky star fields to detect, track and identify RSOs in low-Earth orbits. We aimed to demonstrate the feasibility of using wide FOV, commercial-grade cameras for RSO detection as a first step toward a CubeSat mission The next step in this research is to improve the payload (both hardware design and algorithm development) for space-based RSO observations from LEO, to advance SSA capacities by providing space-based RSO detection data to complement ground-based observations.

## 2. Mission Concept

The RSONAR mission was a technology demonstration of a commercial-grade star tracker for SSA applications. The main research objective was to take images of RSOs from the stratosphere (approximately 40 km in altitude) using a low-resolution, wide FOV star-tracker-like camera. The developed system used a 29.7-degree FOV, contrasting typical systems using 1-degree or smaller FOVs. The mission objective was to progress the technology readiness level of a flight payload that incorporates a commercial-grade star-tracker-like camera. The payload demonstrated in flight that a dual-purpose functionality to image RSOs in addition to the star tracker’s primary purpose of determining spacecraft attitude is achievable. Mission success criteria were defined along three sequential objectives: RSO detections per hour (minimum 1 per hour, ideal 10 per hour), RSO tracking accuracy (minimum 15 arcseconds, ideal 1.5 arcseconds) [21], and RSO identification (minimum 80%, ideal 95%) [22]. 

The RSONAR payload consists of COTS hardware. This includes the camera and lens, the system-on-chip (SoC) development board, the global position system (GPS) sensor, cabling, harnesses, and connectors. The survival of these components would confirm that COTS hardware is suitable for use in payloads for high-altitude missions.

The payload interfaces with CSA’s gondola, designed to operate at an altitude of approximately 40 km, coast for 4 to 8 h, and descend. The duration of the flight is typically about 13 h, for which the payload operates during the night portion of the flight. During the flight, the payload takes images of the sky, varying the camera’s resolution, exposure, and delay depending on the altitude of the balloon, by using altitude values obtained from an onboard GPS to change camera parameters. Operating the payload in this way ensures the largest number of high-quality images to be taken during optimal observation conditions, while the balloon is coasting at the maximum altitude.

## 3. Technology Description

### 3.1. Electronics

The Xilinx PYNQ-Z1 SoC development board (Figure 1), sourced from Digilent, located in Pullman, WA, USA, was used as the onboard computer (OBC). The system consists of a Zynq-7000 SoC, which itself consists of field programmable gate array (FPGA) fabric and a dual-core ARM Cortex-A9 processor [23]. The processor hosts the operating system, which contains the image capture application and driver management, while the FPGA fabric is used to control the pins used for the GPS module. During testing, the board’s ethernet and USB ports were used for debugging purposes. The Arduino headers and PMODA pins were used to interface with other sensors and subsystems. The PYNQ-Z1 board was provided with a Linux-based Operating System (OS), which was used to simplify software design. The captured images were stored in the same partition as the OS. Extensive power-out testing showed that there was no need to mount the OS into its own read-only partition.

A ZED-F9P GNSS module [24], integrated into a SparkFun GPS-RTK2 board, was used to receive GPS signals, providing information such as time, altitude, longitude, and latitude. This module is connected to the PYNQ-Z1 board via its Arduino headers.

An Innodisk industrial-grade 512 GB microSD card [25] was used for onboard storage, as well as to hold the operating system. The microSD card was chosen for its operating and survival temperatures, capable of functioning at −40 °C, which the system could be subjected to in a stratospheric balloon mission. 

### 3.2. Optics

The pco.panda 4.2, sourced from PCO Imaging, located in Kelheim, Germany, a scientific complementary metal-oxide semiconductor (sCMOS) camera [26] was chosen for this mission. This compact monochrome camera is similar in FOV, resolution, and exposure time to a typical star tracker used for attitude determination on satellites. The camera features a maximum resolution of 2048 × 2048 pixels at a variety of exposure times.

Paired with the camera is a wide FOV lens, given the requirement for RSO detection. The lens used is the ZEISS Dimension 2/25, sourced from Carl Zeiss Industrielle Messtechnik GmbH, located in Oberkochen, Germany, having an aperture of 5.7 cm [27]. Combined with the pco.panda 4.2 camera, a full-angle FOV of 29.7° is obtained.

### 3.3. Structure

The payload structure design was based on a CubeSat form factor, a class of miniature satellite with dimensions of 10 cm × 10 cm × 10 cm [28]. A 2U-form factor was implemented, with total dimensions of 10 cm × 10 cm × 20 cm. However, additional triangular prism segments were added to the payload to host a secondary payload for the mission and to provide a 45-degree elevation for viewing geometry. Figure 2 provides a computer-aided design (CAD) model of the RSONAR payload with these additional segments, from multiple perspectives.

The payload itself was mounted to the gondola using a metal interfacing plate. The payload was screwed onto this plate using 10 stainless-steel bolts, nuts, and washers. The assembly was then attached to the gondola using five C-clamps. Figure 3 illustrates the payload and interface plate mounted on the gondola for flight.

The chassis is primarily fabricated from aluminum 6061-T6, while the fasteners are made of AISI 304. The payload’s mass (2.1 kg) falls well under the maximum mass listed by the mission requirements. The interface plate, not included in the mass requirement, has a mass of 2.5 kg alone. Table 1 below shows the masses of the payload components.

To conform to safety requirements, a simulation was performed to examine the payload and interface plate under a variety of load cases, alongside calculations. For all load cases, neither the payload nor the interface plate was expected to detach, given that the stresses were well below the yield strength of the materials. The finite element modeling (FEM) analysis indicated that the highest stress response was at the corner bolts (not on the payload structure itself) where minimum risk is expected.

The payload does not feature an active thermal control (heating or cooling) system, as the electronics were shown to produce enough heat such that this is not a concern, while the upper temperatures are within the limit. A passive thermal control approach was adapted using an aluminum polyamide covering (shown in Figure 3b) as a thermal blanket and reflector, allowing the payload to retain heat during the night and reflect some solar radiation during the day to keep the payload cool. Radiation strike mitigation strategies were not used for this payload.

### 3.4. Communications

To simplify the design, communications to the payload during flight were not implemented. Instead, an autonomous algorithm and onboard storage were used to manage bootup and shutdown, camera settings, payload health, and data storage. In future flights, the National Centre for Space Studies’ (French: Centre national d’études spatiales, CNES) communication subsystem, PASTIS, will be used for telemetry and command, using the Ethernet IP network protocol.

### 3.5. Software

The Xilinx PYNQ Z1 FPGA board was programmed with a headless version of Ubuntu, which makes software development and troubleshooting more convenient. The application driving the camera control was written in C++. Existing header files and functions were used to rapidly develop the application. The application makes use of a GPS receiver to determine the time, altitude, and variation over time, which indicates how stable the payload is and thus what quality could be expected from the images. Therefore, the application was designed to take more pictures at a higher quality during steadier flights.

To achieve this high-quality imaging, a mode scheme was programmed. Each mode corresponds to a mode that was expected to be seen during the flight: ascend, descend, coast, and DnD (dawn and dusk). A safe mode was also programmed to account for potential software errors during flight. To enter each mode, a set of altitude values over a period of time from the GPS are analyzed. If various conditions are met corresponding to that mode, the mode is selected, and images are taken for a short duration, after which the condition checking using the GPS values is repeated.

Ascend mode was created to account for the balloon take-off and other altitude increases, during which time imaging would not be ideal. The resolution is reduced to 512 × 512 pixels, and 20, 10, and 10 images are captured with exposure times of 100 ms, 500 ms, and 1000 ms, respectively. A 10 min delay was added at the end of this mode to further optimize storage used in ascend mode.

Descend mode is similar to ascend mode, created to account for the balloon descent at the end of the mission. The settings are the same as ascend mode. This mode was programmed separately in case changes are desired between ascend mode and descend mode. This was also the mode intended to be used when the balloon is in a powered state but still on the ground.

Coast mode is to account for the condition where the balloon is floating in the stratosphere at a steady altitude. This would be an ideal case for image taking, and thus the image resolution is increased to 1024 × 1024 pixels, and 20, 10, 10, and 10 images are captured with exposure times of 100 ms, 500 ms, 1000 ms, and 5000 ms, respectively. This was done to get a variety of data in this mode. A 10 min delay was also added at the end of this mode.

DnD mode is for image taking during the ideal observation time—the hours preceding dawn and following dusk, during which the light reflected off satellites would be greatest from the perspective of the camera. The resolution in this mode is varied between 2048 × 2048 pixels and 1024 × 1024 pixels, capturing 20 images for each of those resolutions, while the exposure time is set at 100 ms. A delay of just 15 s was added to the end of this mode. The mode was designed such that the most pictures would be taken in this mode.

Lastly, a safe mode was programmed for the event that the GPS data could not be read, invalid, or if a system error is detected. This mode was developed to be as simple as possible, and thus a single resolution and exposure time were used for this mode. The resolution is set to 2048 × 2048 pixels, capturing 10 images with an exposure time of 100 ms. The highest resolution was used to maximize the information that could be acquired. A short exposure time was used over a long exposure time because the short exposure images could be stacked in post-processing to create streaked images for RSO streak detection algorithms, while they could be used individually for RSO point-tracking algorithms. A delay of 20 s was added to the end of this mode.

The software is to be executed on bootup, restarting in the event of a power outage during the mission to ensure that the Linux kernel can properly shut down and reboot. A power-down signal is issued by the team through the CSA’s on-ground power distribution unit (PDU) interface at the end of the mission to safely shut down the payload. Figure 4 outlines the camera operation from bootup.

## 4. Implementation

Figure 5 below outlines the connections between the major components of the payload.

The Xilinx PYNQ Z1 FPGA board connects to the pco.panda 4.2 camera using a USB-A to USB-C cable, which offers sufficient power to the camera as well as data transfer of images. The GPS receiver module connects to the FPGA board via its Arduino headers. The custom sensor board connects to the FPGA board via the PMODA pins. The SD card hosts the operating system, which in turn holds the images, hardware drivers, and sensor interfaces. Power is delivered via the onboard power jack.

Power from the gondola was provided from the PDU, connected via a female PT02E-12-3S connector. The RSONAR payload itself implements the same female connector, thus a male-to-male connector was chosen, ensuring the correct pinouts for the MIL-DTL-26482 standard. Within the payload, the power is routed through two fuses, two tantalum capacitors, and a DC-DC converter, as outlined in Figure 6. The fuses are rated at 3.2 A, derated based on a study recommending fuses to be derated by 50% for vacuum environments [29]. The capacitors are rated at 100 uF and 220 uF, respectively. The DC-DC converter converts incoming power at a different voltage to a steady 12 V output. The PYB20-Q48-S12-DIN from CUI Inc is used to achieve the power regulation, and convert an incoming voltage in the range of 18 V to 75 V down to a voltage of 12 V, and supplying a current of 1.667 A. All components in PDU meet a minimum temperature specification of −40 °C. Figure 6 below outlines the power connectors and components.

## 5. STRATO-SCIENCE 2022 Campaign Overview

The STRATOS stratospheric balloon program aims to provide Earth and space science and engineering research community the opportunity to test newly developed payloads, collect data from a near-space environment, and train next-generation scientists and engineers. During the campaign in August 2022, named STRATO-SCIENCE 2022, four balloons were launched from the Timmins Stratospheric Balloon Base, carrying a variety of payloads. Various flight profiles were available, reaching as high as 40 km in altitude [30].

During integration on-site, GPS data collection from the payload configuration proved difficult, as there was no access to a clear GPS signal with the thermal blanket in place. Consequently, in the absence of a reliable GPS signal, the payload was reconfigured to operate exclusively in safe mode. Safe mode was modified to capture 27 images at a 1024 × 1024 resolution, using the same 100 ms exposure time. The resolution was decreased so that the number of images captured per set could be increased, and so that the delay between sets could be decreased to 4 s. These parameters were chosen to maximize microSD card usage.

The RSONAR payload was manually switched on 30 min before launch. The balloon carrying the payload, depicted in Figure 7, launched at 11:37 PM local time on 21 August 2022, from the Timmins Stratospheric Balloon Base at the Victor M. Power Airport. The payload’s current draw, transmitted from the balloon in real-time, was used as the method of monitoring the status of the payload since expected current draws were previously determined in testing. For the duration of the flight, the current draw was monitored and no abnormal behavior was detected.

The payload was switched off manually at 8:30 AM local time the next day. The balloon continued to fly into the day, collecting valuable near-space data for all other payloads, and landed at 1:10 PM local time.

## 6. Results

### 6.1. Imaging Modes

Given that the payload was forced into safe mode due to an unreliable GPS signal, the algorithm’s mode-switching function could not be observed. However, given the altitudes and changes in altitudes observed through post-mission data, the algorithm would have correctly changed modes to account for the balloon’s ascent, coast, and descent. This was tested by running the algorithm on the GPS data provided by the CSA, creating a script to feed the GPS data to the algorithm at the expected frequency and verifying that the algorithm determined the correct mode to switch to. For future missions, the GPS will be relocated outside of the payload and clamped to the gondola chassis where a clear signal can be received.

### 6.2. Preliminary Image Processing

Over 93,000 images were collected during the active flight by the RSONAR payload. Several algorithms have been developed to detect RSOs from images and return statistics. Detailed analysis of the RSOs detected in the images, both theoretically possible and experimentally determined, will be a part of future work. Preliminary analysis is given below.

#### 6.2.1. RSO Streak Detection

One of the algorithms is an RSO streak detection algorithm, which aims to scan large image databases and find sequences that may contain Resident Space Objects (RSOs) through bulk processing. The algorithm performs bulk processing to preserve time and computational resources. Bulks of sequences are stacked by weighted addition to create a single image. These images are stacked without compensating for star drift, given the minimal movement of stars in these sequences. Image processing techniques such as Wiener filtering, Canny edge detection, and thresholding are applied to the image to reduce noise and detect objects. Smaller objects such as stars are removed by a size threshold and only larger objects such as RSO streaks are left behind. Images that contain a potential object of interest are then considered to be ‘Priority 1′ whereas images with just stars are considered ‘Priority 2′. The algorithm can also detect large obstructions such as Earth’s limb and classifies those images as ‘Priority 3′. Overall, the goal of this algorithm is to reduce the need for manual inspection of databases, which can be tedious and time-consuming. Instead, the classification algorithm provides a snapshot of which image sequences may contain an object of interest. 

Before processing the image set, the first 5400 images were discarded, since the balloon was ascending during this time, causing even stars to streak in the images, resulting in many false positives. From the remaining images, the first 12,130 images were processed as a preliminary test of the algorithm. Since images were captured with a 100 ms exposure time, multiple images in the sequence needed to be stacked together into a single image so that RSOs would appear as streaks. The reduced image set was stacked into sequences of 27 images. Processing the images with the streak detection algorithm returned 51 sequences (with 27 images per sequence) with RSO streaks in them, with an accuracy of 90%. The algorithm was executed again, this time stacking 9 images at a time instead of 27 images to form a single image. This resulted in 69 sequences (with 9 images per sequence) with RSO streaks in them and an accuracy of 98%. Information such as the length of a streak and the brightness of a streak will be implemented in the algorithm in the future. Further analysis such as known and unknown RSO identification (by correlating detections with an RSO catalog, mission objective three) is currently being conducted by developing an IOD algorithm. RSO Figure 8 below shows an example of the resulting image created by stacking images together. Stars appear as stationary light sources, while RSOs appear as streaks in the image. Multiple streaks are visible, corresponding to multiple RSOs.

#### 6.2.2. RSO Point Detection

Another algorithm performs RSO detection and tracking by considering the movement of the RSO points through the images, temporally [31]. The algorithm consists of a convolutional neural network (CNN) combined with a graph-based multiple object tracking (MOT) algorithm. The CNN takes images as input and classifies them as containing detections (RSOs or stars) or not containing detections. The images classified as detections are then used as inputs to the next algorithm, which uses a k-partite graph layout (where k is the total number of images in a sequence, and each detection is a node in the graph). This algorithm distinguishes RSOs from stars by groupings of angular motion, since stars move differently from RSOs.

Preliminary processing of the RSONAR images using this algorithm has found over 500 RSO detections throughout the images. The performance of the algorithm is still being evaluated, and metrics will be provided in the future. Further analysis such as RSO tracking accuracy and RSO identification, mission objectives two and three, respectively, is being conducted, and will also be shown.

The number of RSO detections over the course of the flight has proven a successful mission. Further processing is being carried out to extract data from these images, such as attitude estimation, orbit determination, and optical property estimation. 

### 6.3. Image Delays

The images taken during the mission were timestamped down to the millisecond. Given that the exposure time was consistently 100 ms, it was determined that there was a significant delay between each of the pictures. Ideally, the timestamps would increase in intervals of 100 ms, but are in intervals closer to 400 ms, indicating delays of around 300 ms. Preliminary investigation has shown that the microSD card used for the mission appears to be the primary cause of the delays. Though chosen for its thermal resilience, the microSD card is of UHS Speed Class 1 (U1), which appears to have been too slow for the images. Testing with a UHS Speed Class 3 (U3) card yielded much smaller delays. The team has not found a microSD card with the same or better operating temperature as the one that was used. Given that the microSD card is a critical component, the delays must be managed in a different way, or a faster, limited operating temperature microSD card must be rigorously thermal tested for use. In the future, we plan to use image compression algorithms or different imaging rates and sizes to reduce the data rate.

### 6.4. Thermal Environment

The PCO camera used in the RSONAR payload had three different temperature sensors at various locations within it. These temperature sensors recorded the PCO camera’s internal temperature for the duration that the payload was powered, from 11:00 PM local time to 8:30 AM local time the next day. The gondola itself had a temperature sensor recording the environment temperature for the duration of the flight, from 11:37 PM local time to 1:10 PM local time the next day. Figure 9 below shows the three internal temperatures of the PCO camera plotted with the environmental temperature recorded by the gondola.

During payload development, the cold temperatures were the biggest concern, given that most of the components had minimum operating temperatures of −40 °C. This minimum temperature rating was not low enough to account for the almost –80-°C temperature expected during the ascent through the troposphere. However, all the components of the RSONAR payload survived the mission, and post-mission testing has shown no thermal damage observed in the components.

During the mission itself, the coldest environment temperature observed was −77 °C, while the coldest temperature observed in the PCO camera was −13 °C, well above the camera’s minimum operating temperature. Furthermore, the temperature of the PCO camera decreased at a much smaller rate than the environmental temperature.

However, the maximum temperature range appeared to be an area of larger concern. After sunrise, the PCO camera temperature began to increase at a much larger rate than the environment. At the time that the payload was powered off, the environmental temperature was −30 °C, while the hottest temperature observed in the PCO camera was 55 °C. It appears that this temperature would have kept increasing, given the positive rate of change of the temperature when the payload was shut off. This temperature may have passed the 60-°C maximum operating temperature, which may have caused damage to the components.

From the temperature data, it can be determined that passive heating (the thermal blanket and heat generated by the PCO camera itself) was enough to keep the PCO camera well above its minimum operational temperature during the mission. Given that the components survived with no thermal damage, it can be determined that powering off the payload shortly after sunrise and thus removing the heat generated by the PCO camera, was enough to keep the PCO camera below its maximum operating temperature during the mission.

Given that the temperature sensors were integrated into the camera electronics, they stopped recording the temperature after the camera was shut off. In the future, we will make use of external temperature sensors coupled to a separately powered circuit, such that the sensors can continue to record the temperature even after the camera is shut off. The environmental temperature data provided by the CSA can be used as a temperature profile for thermal testing on the ground, which can provide a way to validate the system for future work.

## 7. Conclusions

The risk of in-orbit collisions is an ever-growing threat, and the need for SSA continues to grow with it. Optical imaging remains a viable method of detecting RSOs, but ground-based telescopes remain expensive, have large integration times, and limited FOVs. 

The RSONAR mission successfully demonstrated the feasibility of using a star-tracker-like, wide FOV camera for imaging RSOs. Over 93,000 images were captured, with preliminary analysis showing over 500 RSOs detected within them. The payload was developed entirely using COTS components. These factors present a cheaper, sub-orbital, rapid imaging alternative to typical ground-based solutions for increasing SSA, demonstrating key technologies to extend the payload to a space-based mission.

Though the cold temperatures were thought to be the biggest concern, the heat generated by the COTS components, as well as the thermal insulation from the aluminum polyamide covering, helped the payload maintain temperatures well within operating specifications. The upper-temperature range was the bigger concern, with device temperatures nearly reaching the maximum operating temperatures before the payload was issued a shutdown signal from the ground.

Given the promising preliminary results, we plan to advance the RSONAR payload for a space mission. However, further changes need to be made to overcome some of the issues faced, and to ensure the design is ready for the challenges of a low earth orbit environment. We aim to use a radiation-hardened version of the Zynq-7000 SoC, combined with space-grade sensors and custom pins to ensure the hardware is ready for the space environment.

For future work, we will demonstrate increased onboard image processing to detect RSOs in real-time. This involves keeping only images containing RSOs to reduce the amount of data stored and downlinked, as well as the time spent processing images for RSOs on the ground. Furthermore, real-time image processing for RSO detections in future stratospheric missions will serve as a technology demonstration for real-time RSO detection in space-based missions. This is highly desirable because it is the first step in enabling autonomous, real-time action, such as RSO tracking, changing camera parameters for better RSO imaging, and performing satellite maneuvers, without needing a human in the loop. The FPGA environment will be better leveraged to accomplish this by offloading processing-heavy tasks to the FPGA fabric, and sequential tasks and sensor-interfacing to the integrated ARM processor.

## Figures and Tables

**Figure 1 sensors-23-06595-f001:**
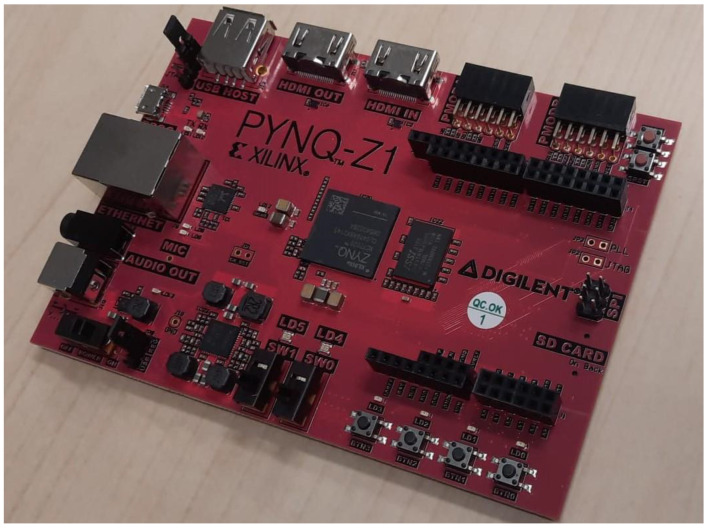
Xilinx PYNQ-Z1 FPGA development board.

**Figure 2 sensors-23-06595-f002:**
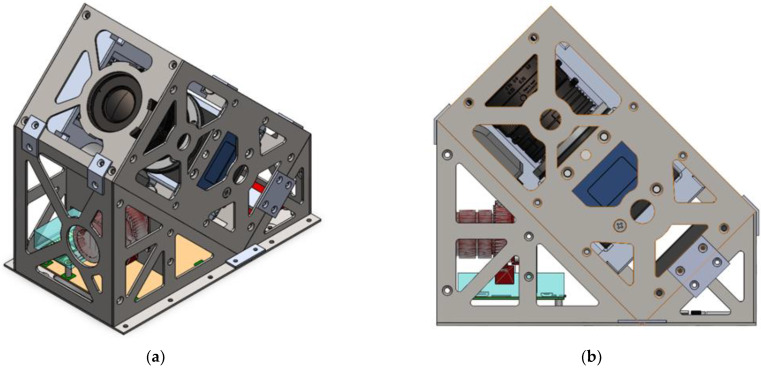
RSONAR payload CAD model outlining the structure and some electronics within the payload from two views: (**a**) Isometric view; (**b**) Side view. Note that triangular prism-like segments are added to the pain 2U segment.

**Figure 3 sensors-23-06595-f003:**
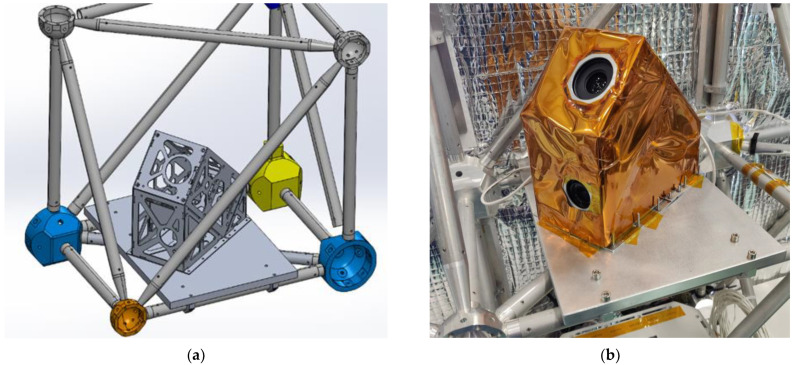
RSONAR payload fastened to the gondola. (**a**) CAD model depiction of the mounting scheme; (**b**) Actual integration of the payload to the gondola.

**Figure 4 sensors-23-06595-f004:**
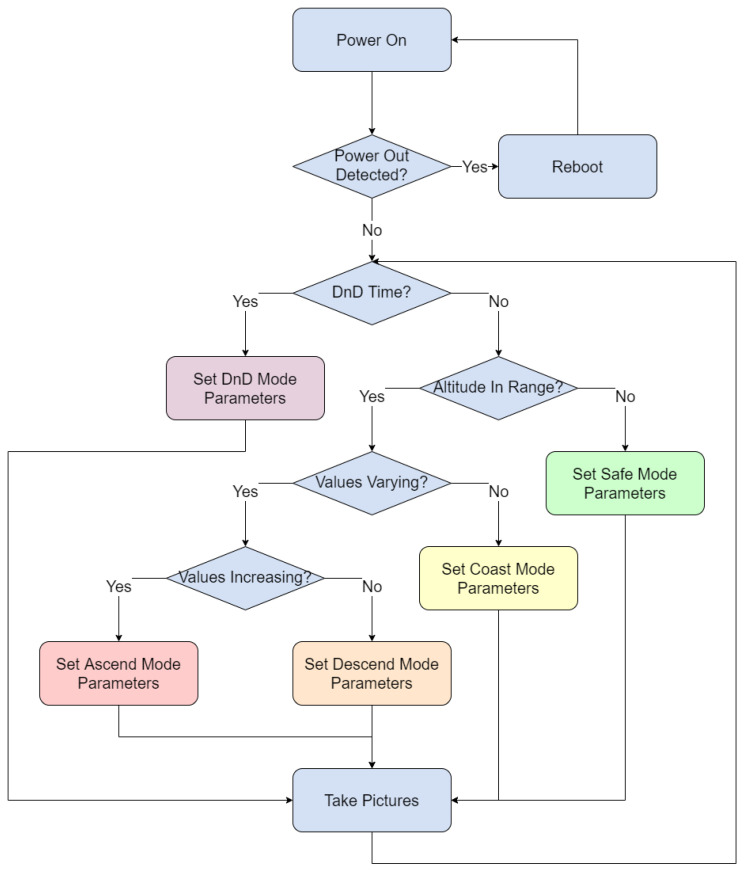
Block diagram outlining the autonomous algorithm used to power on the payload, check the altitude and time, and take images with pre-defined camera parameters corresponding to the mode.

**Figure 5 sensors-23-06595-f005:**
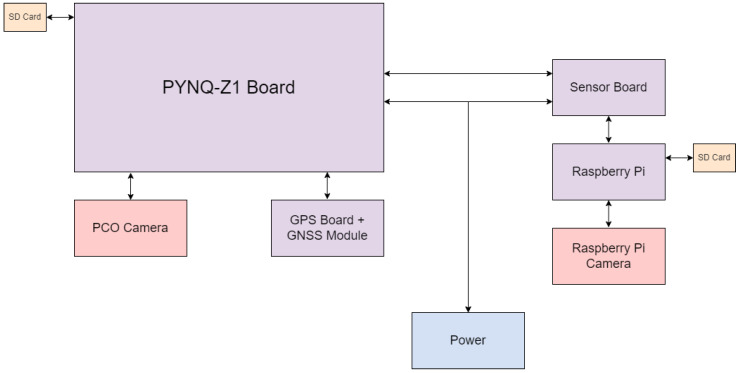
Diagram outlining the connections between the electronics in the RSONAR payload.

**Figure 6 sensors-23-06595-f006:**
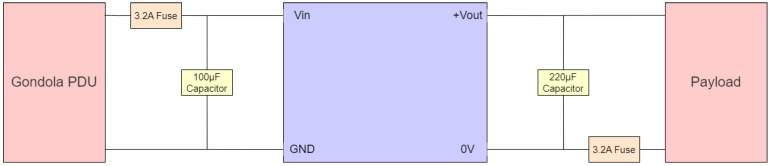
Diagram outlining the electrical components used on the power distribution unit.

**Figure 7 sensors-23-06595-f007:**
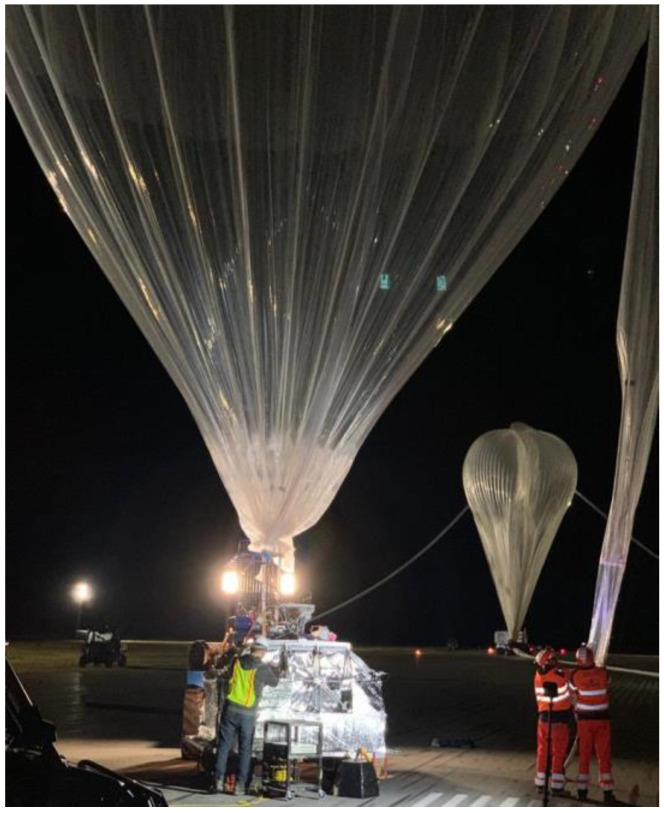
Pictured in the foreground is a smaller balloon used to keep the attached gondola steady upon launch. In the background is the larger balloon being inflated to launch. The larger balloon expanded even further as it ascended into the stratosphere.

**Figure 8 sensors-23-06595-f008:**
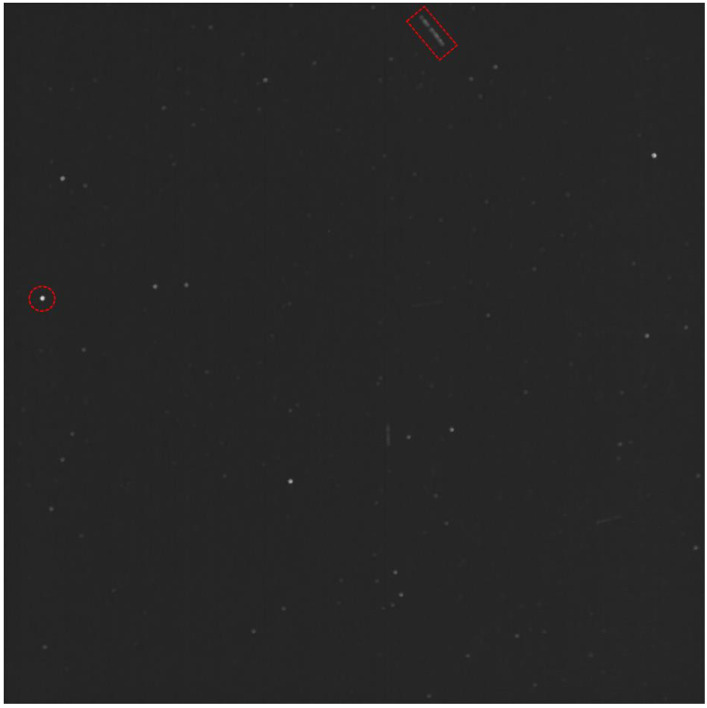
Example of an image created from stacking a sequence of images. Stars appear as bright dots (example contained in dashed red circle), while RSOs appear as streaks (example contained in dashed red box). Multiple streaks are visible in the image, corresponding to multiple RSOs.

**Figure 9 sensors-23-06595-f009:**
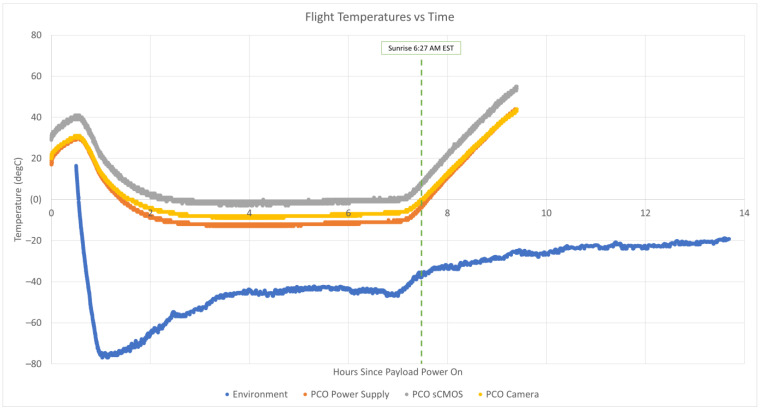
Plot of the temperatures inside the PCO camera at various locations (sCMOS sensor, camera, and power supply) for the duration that the payload was powered alongside the temperature of the environment for the duration of the flight. Temperature logging of the camera ended at 8:30 AM local time, while environmental temperature logging continued until 1:10 PM local time. This work is based on observations with the CNES temperature sensor under a balloon operated by CNES, within STRATO-SCIENCE 2022 and in the framework of the CNES/CSA Agreement.

**Table 1 sensors-23-06595-t001:** Components of the RSONAR payload and their corresponding masses.

Item	Mass (g)
Fasteners (screws, nuts, washers)	80
Aluminum chassis	780
PCB and breakout boards	230
Optics and antenna	955
Wires and cables	100
Total	2145

## Data Availability

The data presented in this study are available on request from the corresponding author. The data are not publicly available due to continuing research and containing sensitive data.

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
