# Peer review of "Stratospheric Night Sky Imaging Payload for Space Situational Awareness (SSA)"

_sensors, 2023, doi:10.3390/s23146595_

Round 1

Reviewer 1 Report

This paper described a prototype image system for SSA.  It gave a detailed design description. This system successfully passed the trial flight with promising results.  The development of such a system will be valuable for the tasks of monitoring the ever-increasingly crowded space instrument development.

Some comments

It would be good to explain a bit more why the ‘Active tracking is suitable for GEO objects, however the tracking requirements for a LEO observation introduce complications on the tracking system.” 

 The GPS fault was attributed to the thermal blanket.  There is no mention of how to mitigate this problem.

From the post data processing, it was found that without the GPS, “the algorithm would have correctly changed modes”.  Can it be elaborated a bit more?

The temperature of the sensor raised some concerns but seemed may not be a big problem.   However, it is a pity that the sensor temperature monitoring was switched off at the same time the system was switched off.  Moreover, the system that survived once may not indicate that it can survive multiple uses.  From the temperature point of view, it would not be difficult to simulate the same environment on the ground to test this. 

 Could the data taken from the trial test be compared with the existing ground monitoring system to validate it?

Reviewer 2 Report

The authors present a very interesting experiment that uses cheap off-the-shelf hardware for detecting resident space objects (or space junk).

The authors describe their experimental setup in great detail that would allow to replicate the experiment. They also describe the course of the experimental campaign very well. Overall the article is written in  a clear and concise manner which is easy to follow and to understand.

What I miss is a more detailed result analysis of the captured data. In the introduction three objectives were defined but only one of it is analyzed (RSO identification). The conclusion also mentioned that over 500 RSOs were detected but there is no corresponding analysis in the result section. How many of the RSOs are known? How many are new? How many could have been detected theoretically? What are typical miss-detections? How do the objectives change over the course of the experiment (are there more miss-detection at the beginning or end? or things like this).

There are also a few conceptual questions that remain after reading the paper:

* What are typical narrow FOV when doing RSO detection? How does it compare to the presented setup?

* Radiation does not seem to be an issue. At least it is not mentioned anywhere except when discussion future steps to move the space environment.

* Is the operating systems partition mounted read-only to prevent memory corruption in case of power outages? Images could be stored in a separate partition mounted read/write.

* Why does the processing need to be done in real-time on the board? Lossless compressed images of that kind should require only very minimal memory.

* How many images are captured in sequence during a mode, i.e. before delay kicks in and the sensor data is analyzed to decide for the next mode?

* Why are 100ms images stacked instead of using longer exposure times? I mean apart from the fact that only the emergency mode was triggered due to GPS issues.

* How has the RSO accuracy been determined? Manual inspection and labeling of some of the images?

* Are the images stacked as they are or are they shifted before to compensate for drift? For example using auto-correlation as most of the image should show stars which probably are considered fixed in position in such short time frames.

* What are the planned countermeasures to the GPS issues?

Some further suggestions:

* Maybe you can consider automotive cameras/chips for similar future setups. They are also quite cheap and have higher temperature specs (typical -40 to 100°C) because they are mounted behind the windshield with a black cover to prevent stray light which heats up the setup quite a lot in sunny regions.

* You found some image delays. I was wondering how you actually captured the images. If you use video4linux you need to make sure to set the image capture buffer to zero. Otherwise, you will get old images if your pipeline is slower then the frame rate because video4linux always returns the oldest image from the buffer which means you will always get 300ms old images in case you have a buffer size of three and your pipeline runs with less then ~3Hz.

Some minor comment on the layout. Several images are broader than the text width.

Reviewer 3 Report

Minor editing of English language required.

Round 2

Reviewer 2 Report

Thank you very much for the thorough handling of the questions. The document has improved to a substantial extend.

Some suggestion for future problem mitigation:

I read that the SD card writing speed caused the issue with the delays. One solution we used in a robotics project was to store the images into the RAM first (file pointers directly into memory works, so you can use normal image writing API) using a queue and having a separate process for writing to the SD card which takes the images from the RAM queue. Of course this only works if there is some time between recording bursts where the queue can be emptied but as far as I understood your recording setup this could work (depending on RAM size and size of the images recorded in burst). This can also be combined with the planned compression.

Reviewer 3 Report

I read the new version of the manuscript and I think that it has been significantly improved. The Authors have satisfactorily responsed to all my previous comments. I fthink that it can be considered for publication in Sensors.

Minor editing of English language required